# The Role of Inverted Papilloma Surgical Removal for Sleep Apnea Treatment Success—A Case Report

**DOI:** 10.3390/medicina59030444

**Published:** 2023-02-23

**Authors:** Ana Maria Vlad, Cristian Dragos Stefanescu, Catalina Voiosu, Razvan Hainarosie

**Affiliations:** 1Institute of Phonoaudiology and Functional ENT Surgery, 21st Mihail Cioranu Street, 050751 Bucharest, Romania; 2Faculty of Medicine, “Carol Davila” University of Medicine and Pharmacy, 37th Dionisie Lupu Street, 020022 Bucharest, Romania

**Keywords:** sleep apnea, nasal surgery, inverted papilloma, CPAP, multimodal treatment, COPD

## Abstract

In recent years, increased attention has been directed to sleep apnea syndrome due to its high prevalence and preventable severe health consequences. Besides enhancing the risk of cardiovascular, cerebrovascular, and metabolic disorders, it determines increased daytime somnolence, cognitive impairment, and delayed reaction time. These symptoms, determined by sleep fragmentation and chronic hypoxemia, can result in a decrease in professional performance and, moreover, could have tragic implications, especially in patients with high-risk professions. We present the case of a 58-year-old male-truck driver, known to suffer from uncontrolled OSA and chronic obstructive pulmonary disease, who presented to our ENT department for incapacitating daytime somnolence and severe nasal obstruction. These symptoms were caused by a voluminous sinonasal inverted papilloma, occupying the entire left cavity with extension in the nasopharynx. Following nose permeabilization, the patients’ APAP compliance grew substantially, with a dramatic decrease in daytime sleepiness and improvement in polysomnographic parameters. Due to the overlap syndrome of OSA and COPD, an oxygen supplementation was added to PAP therapy by a pulmonologist, improving pulse-oximetry parameters and resulting in the best outcome for the patient. Through this case report, we aim to emphasize the importance of multimodal, personalized treatment of sleep apnea with a focus on nasal surgical permeabilization. At the same time, we sustain a multidisciplinary approach, especially in patients with sleep apnea and associated pathologies, to obtain therapeutic success. We propose increased attention to the early recognition and proper treatment of sleep apnea in patients with high-risk professions as it prevents catastrophes.

## 1. Introduction

Sleep apnea is a multifactorial disease that determines the partial or total obstruction of the upper airway during sleep. Left untreated, it results in important health consequences such as cardiovascular, cerebrovascular, and metabolic disorders. Through micro awakenings and oxygen desaturation characterizing the pathology, it leads to daytime sleepiness, cognitive impairment, psychological problems, and a significant reduction in quality of life. Secondary to the patients’ delayed reaction time, especially in those suffering from frequent desaturations during sleep, OSA can result in tragic traffic road accidents [1]. Inverted papilloma is a rare benign sinonasal tumor characterized by a strong potential for local destruction, a high rate of recurrence, and a risk of malignant transformation. Nasal obstruction can be the main symptom, especially in voluminous tumors. The treatment of choice nowadays is surgical excision via an endoscopic nasosinusal approach with a favorable outcome [2].

In recent years, a multimodality approach has been studied and applied in the integrative management of obstructive sleep apnea. The role of nasal breathing during the night in healthy individuals is well-established. The implications of nasal obstruction on sleep apnea have been addressed in several studies. The switch to mouth breathing secondary to nasal blockage is associated with increased OSA severity and worse oximetric variables [3]. However, isolated nose surgery for sleep apnea improvement is controversial. Still, nasal surgery significantly improves subjective symptoms of daytime somnolence and is considered to have an important role in CPAP compliance and efficiency [4]. This is of great importance in enhancing the quality of life and results in an indirect reduction of health consequences through correct CPAP utilization. In some cases, in the same patient, sleep apnea and chronic obstructive pulmonary disease (CPOD) can coexist, resulting in so-called overlap syndrome, which brings some particularities in diagnosis and treatment [5].

In this paper, we present the case of a patient known to have sleep apnea and COPD who presented at the hospital with severe daytime sleepiness and nasal obstruction determined by a voluminous inverted papilloma. This is a representative case of individualized care requirements for sleep apnea. Through our case report, we demonstrate that a multimodal, multidisciplinary approach results in successful OSA outcomes and a significant increase in patients’ quality of life.

## 2. Case Report

A 58-year-old patient, known to have sleep apnea and COPD, was referred to our ENT department for a 2-year history of persistent left nasal obstruction, mouth breathing, snoring, daytime somnolence, and gradually worsening sleep apnea episodes. His past medical history consists of an inverted papilloma diagnosed ten years anteriorly, for which he underwent surgery. No recurrences were observed after the end of the treatment.

The patient experienced significant daytime sleepiness that interfered with their daily life. They scored 24 on the Epworth Sleepiness Scale, and reported instances of falling asleep while driving as a truck driver. Hospital admission was sought for further evaluation and management. The flexible nasopharyngoscopic examination revealed the obstruction site. A voluminous papillomatous, yellowish tumor entirely blocked the left nasal fossa extending toward the ipsi- and contralateral choana. (Figure 1). An anterior left nasal deviation was also noted. Mucopurulent secretions were visualized in both nasal fossae.

A computed tomographic scanning of the head and paranasal sinuses was performed, showing extensive parafluid-solid densities completely occupying the left maxillar, ethmoidal, sphenoidal sinuses and the entire left nasal cavity, expanding in the nasopharynx through the left choana. (Figure 2).

A polysomnographic study showed severe obstructive sleep apnea syndrome (AHI = 67.8) with the most prolonged apnea episode duration of 1 min and 55 s (Table 1). Most episodes were represented by obstructive apneas (oA = 73%), while central and mixed apneas represented 1% and 12%, respectively. The rest of the 13% were hypopneas ([Fig medicina-59-00444-ch001]).

These frequent and prolonged apnea episodes influenced pulsoxymetric parameters ([Fig medicina-59-00444-ch002]), determining a desaturation index of 90.0, with a mean O2 saturation of 74% and a minimum saturation of 50% during sleep (Table 2), ([Fig medicina-59-00444-ch003]). The SpO2 reading on the pulse oximeter was 86% during the day, with a pulse range of 65 to 111.

Blood tests were performed, revealing modified complete blood count parameters: hemoglobin 17.9 g/dL (N = 13.1–17), hematocrit 56.2%(N = 39–50%), MCV 103.5 fL (N = 81–101), and RDW 15.7% (N = 11.5–14.5%).

With the patient’s informed consent, surgical intervention was proposed. Under general anesthesia with orotracheal intubation, sinonasal videoendoscopic surgery was performed. It consisted of the excision of the tumoral mass from the nasal cavity and nasopharynx with subsequently left maxillary sinus antrostomy, anterior ethmoidectomy, and sphenoidotomy. (Figure 3) Bioptic fragments from the significant sites of tumor extension were sent to the anatomopathological laboratory for histopathological examination.

The postoperative care consisted of intravenous antibiotics, hemostatic agents, and analgesics for pain management. The patient had a favorable result without postoperative complications. The histopathological result was suggestive of inverted papilloma.

Postoperatively, the symptoms of nasal obstruction and daytime somnolence diminished substantially. Even if the leading cause of the patient’s complaints had been eliminated, further steps were made to manage the case better. Considering that the patient had multi-level obstruction determining sleep apnea (anteroposterior diminished pharyngeal diameter, elongated uvula, increased neck circumference), an APAP mask was recommended.

The polysomnography showed a residual AHI of only 5.4/h in the first month after surgery and PAP therapy prescription. However, even if considerably reduced, pulse oximetry parameters did not reach the levels of desired therapeutic success. APAP treatment was completed with oxygen supplementation during sleep to better manage desaturation episodes. This measure resulted in a very favorable outcome with a ODI of only 18.6 (vs. 90 at first presentation), mean O2 saturation of 91%, and a minimum saturation of 76% (Table 3).

Daytime SaO2 increased from 84% before surgery to 93% ten days after surgery. (Figure 4) Secondary to nocturnal oxygen supplementation, the daytime SaO2 reached 96%.

Secondary polycythemia, highlighted by increased hemoglobin and hematocrit, has been rectified after treatment implementation. Following three months under therapeutic measures, hematocrit decreased by almost 10% (from 56.2 to 47.2%) and hemoglobin by approximately 2 g/d (from 17.9 to 16 g/dL).

A videoendoscopic examination of the nasal cavity was repeated three months after the surgical intervention. The nose was permeable, with no recurrence of nasal tumor observed (Figure 5).

The patient completed the Epworth daytime sleepiness scale revealing a score of 0 points, compared to a score of 24 before the treatment. He described that his quality of life improved significantly.

## 3. Discussion

Obstructive sleep apnea (OSA) is a pathology characterized by recurrent episodes of airway collapse, determining upper airway obstruction during sleep. This leads to subsequent oxygen desaturation, chronic intermittent hypoxia, and repetitive micro awakening, resulting in daytime sleepiness and a wide range of health problems.

The prompt treatment of sleep apnea in the particular case of our patient had a special significance considering his profession. As a truck driver, he represented a real risk for the population. It has been demonstrated that sleep fragmentation and intermittent hypoxemia, characterizing sleep apnea episodes, leads to excessive daytime sleepiness, cognitive processing, and reaction time delays. The most recent and most extensive study on this subject, conducted in Denmark, including all citizens diagnosed with OSA from 1995 to 2015, concluded that sleep apnea patients are not only at an increased risk for road traffic accidents, but also tend to be involved in more severe accidents [1].

We can state that the nasal obstruction determined by the nasal tumor and subsequent severe daytime sleepiness were the main factors that motivated the patient to come to the hospital. Even though he was diagnosed with sleep apnea ten years anteriorly, he chose not to follow any treatment because he did not have such disturbing symptoms as those produced by the tumor. At the same time, he decided not to use APAP therapy even after the onset of the symptoms because of the discomfort created by the high nose resistance.

Nasal breathing is the preferred breathing route during sleep, and various pathophysiological mechanisms could explain the implication of nose obstruction on sleep-disordered breathing and OSA. Firstly, conforming to the Starling resistor model, increased nasal resistance might amplify the collapsibility of oropharyngeal soft tissue due to negative pressure downstream. At the same time, a blocked nose might determine a compensatory switch to mouth breathing that is unfavorable during sleep due to the narrowing of the pharyngeal space, secondary to the base of tongue collapse. Additionally, in the absence of nasal airflow, the receptors for nasal respiratory reflex are not activated, resulting in decreased muscle tone and ventilation. Moreover, a significant quantity of NO is produced in the nose; bypassing it, reduced NO is transferred to the lungs, with subsequent poor blood oxygenation [6,7,8,9].

Several studies approached the effect of isolated nasal surgery on sleep apnea improvement. However, in most articles, there was no statistically relevant improvement in the objective parameters quantifiable by polysomnography. The most recently published systematic review (Schoustra et al. 2022) concluded that the apnea-hypopnea index (AHI) did not improve substantially after isolated nasal surgery in most studies. Yet, it had an impressive effect on daytime sleepiness and overall quality of life [10]. Nonetheless, the authors observed a slight decrease in AHI in the patients followed for more than three months postoperatively and suggested that future research papers should focus more on the extended follow-up to quantify the changes in polysomnographic indices better.

In our case, it is hard to measure the exact impact of surgery on daytime sleepi-ness due to the patient using CPAP immediately after tumor removal. However, the combination of nasal surgery and CPAP use led to a significant improvement in the patient’s Epworth Sleeping Scale score, reducing it from 24 to 0. There is limited re-search on the relationship between inverted papilloma and sleep apnea, but several studies have explored the connection between other causes of nasal obstruction and OSA. A clinical trial published in 2019 [11]. concluded that endoscopic sinus surgery for nasal polyps improves sleep efficiency by dramatically decreasing the arousal index on polysomnography following nasal permeabilization. Through this mechanism, a significant reduction of daytime somnolence resulted.

It is interesting to note the discordance between the subjective and objective parameters change following nasal surgery. Iwata et al. 2020 came up with an intriguing explanation for this phenomenon [12]. They assume that besides lowering daytime sleepiness, the comfortable and profound sleep following nasal permeabilization determines greater collapsibility of pharyngeal structures. Thus, the two processes cancel each other out, resulting in an unchanged AHI.

In agreement with previous studies, a review published in 2022 regarding the holistic care of patients with sleep apnea concluded that restoring nasal breathing is indispensable in optimizing OSA treatment outcomes [4]. This statement is sustained by our case, too. We used a personalized, multimodal treatment for our patient, with great success in improving both subjective and objective quantifiable parameters of OSA. After endoscopic nasal surgery for tumor removal, an APAP mask was recommended. The patient was very compliant to night-time ventilation therapy, with a mean usage time of 87% three months after the surgery. This high compliance is consistent with various studies in the literature sustaining the importance of nasal surgery for CPAP adherence. Brimioulle et al., 2022, reviewed the literature to emphasize the bidirectional link between nose and CPAP use. Even if CPAP’s effects on the nose remain uncertain, all studies reported a correlation between lower nasal resistance and higher CPAP compliance [13].

The overlap syndrome describes the coexistence of obstructive sleep apnea in patients with chronic obstructive pulmonary disease (COPD) [14]. It has been shown that overlap patients are particularly prone to oxygen desaturation during sleep, making the overnight oxygen desaturation frequency (ODI) the most crucial variable to evaluate in this category of patients [5]. Our patient’s polysomnographic indices show extremely frequent desaturations (ODI = 90), supporting the data in the literature. PAP remains the standard treatment for OSA and is also accepted for overlap syndrome. However, in some cases, CPAP alone may not be sufficient for hypoxemia correction, and supplemental oxygen might be needed [15]. Even though the CPAP therapy reduced the oxygen desaturation index by more than half, it was not sufficient for therapeutic success in our case. With 4 L of supplemental oxygen, the measured parameters improved substantially, resulting in an ODI of only 18.6 and a mean oxygen saturation of 91%. These therapeutic measures positively impacted blood test results resolving secondary polycythemia. Hemoglobin and hematocrit reached normal values three months after treatment implementation. Considering the known association between elevated hematocrit levels and increased cardiovascular and all-cause mortality [16], the reduction of almost 10% in our case (from 56.2% to 47.2%) may translate into potentially favorable clinical outcomes.

## 4. Conclusions

Daytime somnolence determined by sleep apnea can result in cognitive function deterioration followed by a decline in professional performance and, in the case of high-risk professions (driver, locomotive mechanic, pilot, etc.), can lead to major transportation disasters.

Nasal surgery facilitates non-invasive pressure support because the patients require lower pressures for similar effects. This implies an improvement in PAP therapy compliance with consequent benefits for the patient, for society (by reducing accidents), and for the health system (by reducing the costs associated with chronic preventable conditions).

The interdisciplinary (ENT, pneumology) and multimodal (nasal surgery, non-invasive ventilation) approaches represent an adequate therapeutic solution for patients suffering from sleep apnea with multiple comorbidities.

## Data Availability

Not applicable.

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
