# Peer review of "The Role of Inverted Papilloma Surgical Removal for Sleep Apnea Treatment Success—A Case Report"

_medicina, 2023, doi:10.3390/medicina59030444_

Round 1

Reviewer 1 Report

Esteemed authors,

 It is a very interesting manuscript presenting a complex case with sleep apnea requiring multidisciplinary diagnosis.

However, there are some aspects requiring little changes:

On Page 4 Line 101 The Table Respiratory evaluation needs numbering and title according to MDPI standards.

On Page 4 Line 102 The Chart detailing the percentage of episodes needs a Title, numbering, and legend according to MDPI standards.

On Page 4 Line 107 The Chart detailing the SpO2 needs title and numbering according to MDPI standards.

On Page 4 Line 108 The Table Evaluation of SpO2/ Pulse needs title and numbering according to MDPI standards.

On Page 4 Line 109 The Chart entitled SpO2 distribution needs description and numbering according to MDPI standards.

On Line 112 instead of “blood tests were run out” use blood tests were performed.

On Line 135 instead of “1 month after surgery” use in the first month after surgery.

On Page 6 Line 141 The table describing comparative values for oxygen supplementation needs numbering and title according to MDPI standards.

Mention Figures 3, 4 and 5 in the text.

On Page 7 Line 151, better transform the table into a text paragraph.

In the discussion section compare the characteristics of the case with more recent articles from the MDPI platform such as Neagos A, Dumitru M, Neagos CM, Mitroi M, Vrinceanu D. Correlations between Morphology, the Functional Properties of Upper Airways, and the Severity of Sleep Apnea. J Clin Med. 2022 Sep 12;11(18):5347. doi: 10.3390/jcm11185347. PMID: 36142994; PMCID: PMC9502432.

At the end of the manuscript please add the sections: acknowledgements, funding, author contribution, institutional review board, informed consent statement, data availability, conflict of interests, according to MDPI instructions for authors.

I look forward to receiving an improved version of the manuscript.

Reviewer 2 Report

1.    Through a case report, the article emphasizes the importance of multimodal, personalized treatment of sleep apnea with a focus on nasal surgery.

2.    However, as a case report, the article is to long, so it is suggested to simplify and highlight the key points.

3.    Can the author provide more clear nasal endoscopic photos?

Reviewer 3 Report

Dear Authors, 

Congratulations for the work done.

Undoubtedly, it is a very adequate work of data collection and presentation quality. The conclusions obtained through this case report, however, are already well known: that OSA should be evaluated in a multidisciplinary manner, including Pneumology, Otolaryngology, Endocrinology and Maxillofacial Surgery. 

Therefore, despite the quality of the article, I believe that it does not provide specific value.
